# Iron Oxide Nanoparticle-Based Ferro-Nanofluids for Advanced Technological Applications

**DOI:** 10.3390/molecules27227931

**Published:** 2022-11-16

**Authors:** Mohd Imran, Anis Ahmad Chaudhary, Shahzad Ahmed, Md. Mottahir Alam, Afzal Khan, Nasser Zouli, Jabir Hakami, Hassan Ahmad Rudayni, Salah-Ud-Din Khan

**Affiliations:** 1Department of Chemical Engineering, Faculty of Engineering, Jazan University, P.O. Box 706, Jazan 45142, Saudi Arabia; 2Department of Biology, College of Science, Imam Mohammad Ibn Saud Islamic University (IMSIU), Riyadh 11623, Saudi Arabia; 3Department of Metallurgical and Materials Engineering, Indian Institute of Technology Jodhpur, Jodhpur 342037, Rajasthan, India; 4Department of Electrical & Computer Engineering, Faculty of Engineering, King Abdulaziz University, P.O. Box 80204, Jeddah 21589, Saudi Arabia; 5State Key Laboratory of Silicon Materials, School of Materials Science and Engineering, Zhejiang University, Hangzhou 310027, China; 6Department of Physics, College of Science, Jazan University, P.O. Box 114, Jazan 45142, Saudi Arabia; 7Department of Biochemistry, College of Medicine, Imam Mohammad Ibn Saud Islamic University (IMSIU), Riyadh 11432, Saudi Arabia

**Keywords:** iron oxide nanoparticles (IONs), ferro-nanofluids (FNs), mass transfer, sensors and actuators, MEMS, coolants, environmental remediation, air and water purifications, magneto-rheology, dampers and sealings

## Abstract

Iron oxide nanoparticle (ION)-based ferro-nanofluids (FNs) have been used for different technological applications owing to their excellent magneto-rheological properties. A comprehensive overview of the current advancement of FNs based on IONs for various engineering applications is unquestionably necessary. Hence, in this review article, various important advanced technological applications of ION-based FNs concerning different engineering fields are critically summarized. The chemical engineering applications are mainly focused on mass transfer processes. Similarly, the electrical and electronics engineering applications are mainly focused on magnetic field sensors, FN-based temperature sensors and tilt sensors, microelectromechanical systems (MEMS) and on-chip components, actuators, and cooling for electronic devices and photovoltaic thermal systems. On the other hand, environmental engineering applications encompass water and air purification. Moreover, mechanical engineering or magneto-rheological applications include dampers and sealings. This review article provides up-to-date information related to the technological advancements and emerging trends in ION-based FN research concerning various engineering fields, as well as discusses the challenges and future perspectives.

## 1. Introduction

Magnetic nanoparticles (MNPs) are nanoscale substances with distinctive magnetic characteristics, which have been extensively employed in a variety of sectors [1,2,3,4]. The rapid advancement and an unprecedented number of studies have elevated MNPs to the forefront of nanoscience and nanobiotechnology [5,6,7,8]. Certain difficulties in manufacturing monodisperse magnetic nanostructures, such as dipolar interactions, particle surface effects, and size controlling, are of major importance. However, new chemical synthesis techniques have made it simpler to limit the nucleation and proliferation of such MNPs. MNPs comprise either several metallic substances or their magnetic oxides and composites [9]. Due to its great biocompatibility, high surface area, and low toxicity, superparamagnetic magnetite (Fe_3_O_4_) is the most popular iron oxide or magnetic oxide [10,11,12]. Additionally, magnetite (Fe_3_O_4_), hematite (α-Fe_2_O_3_), and maghemite (γ-Fe_2_O_3_) are the three greatest prevalent iron oxides found naturally. Such oxides are highly essential in the domain of science and technology [13,14]. Moreover, for particular purposes, magnetic moment, adsorption kinetics, and superparamagnetism can be customized throughout the manufacturing process [15]. Superparamagnetic NPs including IONs have been extensively studied among the numerous kinds of NPs since they do not preserve any magnetization after the magnetic field is withdrawn [10,11,12,13,14]. Because of their ease of functionalization with polymers and other materials, IONs (Fe_3_O_4_ and Fe_2_O_3_) have indeed been extensively utilized for in vitro diagnostics among other purposes [13,14].

IONs can be synthesized using a variety of approaches, such as wet chemistry or “bottom-up” pathways including sol–gel, electrochemical, hydrothermal, co-precipitation, flow injection syntheses, solvothermal, reverse micelles, and laser pyrolysis techniques [3,7,8,12,13,14,16].Organic liquids or water are commonly employed as base fluids in the synthesis of ION-based FNs to be utilized for different technological applications including sensors [13,17,18,19,20,21,22,23,24,25]. Similarly, in this demanding and challenging time, it is necessary to emphasize the advancement of intelligent sensor systems combined with Internet of things (IoTs), fifth-generation (5G) connectivity, artificial intelligence (AI), and machine learning (ML) techniques, which are evolving in response to the demands of future generations [26,27,28,29,30]. Moreover, porous structures can also be used in the sensing process to increase the sensitivity of the signal output utilizing an electrochemical method [31,32,33,34,35]. Figure 1 depicts a number of prospective advanced technological applications of ION-based FNs concerning different engineering fields.

Several studies on the preparations, characteristics, and biomedical applications of ION-based FNs have already been documented in recent years. Therefore, the primary goal of this review article is to highlight current advancements in ION-based FN development and applicability in different engineering fields. This review article starts with the introduction of MNPs, IONs, and their corresponding FNs, as well as their processability and potential applications. Following that, it is divided into four sections: the first includes chemical engineering applications such as mass transfer processes; the second section encompasses electrical and electronics applications such as magnetic field sensors, FN-based temperature sensors, FN-based tilt sensors, MEMS and on-chip components using FN, actuators, and cooling for electronic devices and photovoltaic thermal systems; the third section contains environmental engineering applications (water and air purifications); the fourth section briefly discusses magneto-rheological applications (dampers and sealings). Lastly, conclusion, challenges, and future perspectives of IONs-based FNs are discussed.

## 2. Chemical Engineering Applications

Chemical engineering is one of the important branches of engineering disciplines where a number of processes are used from the laboratory to large-scale industrial applications. Chemical engineering involves the use of materials and energy from the molecular level to the large-scale industrial level, transporting and transforming them. Various processes such as mass transfer, heat transfer, pressure drop, plant design, operations, chemical reaction engineering, modeling and simulation, and chemical and biological analysis are also apart of chemical engineering. Some of the important processes where ION-based FNs are utilized for their improvement are discussed below, which makes them useful for various industrial applications.

### Mass Transfer

Mass transfer is an interesting phenomenon in chemical engineering that involves the movement of mass or chemical species from a region of higher concentration to a region of lower concentration. Mass transfer is used in various industrial applications and commonly occurs in liquid extraction, absorption, distillation, evaporation, drying, precipitation, and membrane filtration. Generally, a mass transfer may take place within two liquid phases, between a liquid and a solid phase, or between a gas and a liquid phase. One such interface is the gas–liquid interface, which is utilized in most industrial processes. Gas–liquid mass transfer has applications in wastewater treatment, industrial chemical reactions, and bio processing [36]. Magnetic NP-based ferrofluids may control the mass and fluidity of the carrier liquid such as ethylene glycol, oil, and water. The electromagnetic field affects the gas–liquid interface as it is sensitive to the magnetic field and guarantees the required regime for the mass transfer processes, which is decided by the residence time and surface area [37].

Suresh et al. studied the rate of mass transfer which was enhanced using FNs in a gas–liquid interface region and oscillated in the presence of a magnetic field [38]. They obtained a 40–50% increase in the rate of mass transfer, which was found to be dependent on the loading of FNs and strength of the magnetic field. Komati et al. also studied the rate of mass transfer during CO_2_ absorption in an amine solution while adding the FNs in the liquid phase [39]. It was observed that, upon the addition of ION (Fe_3_O_4_)-based FNs in the liquid phase, the rate of mass transfer increased to 92.8%. A similar study was conducted by the same group, where they investigated the effect of IONs (Fe_3_O_4_) in the gas–liquid phase on the mass transfer rate [40]. The enhancement of the mass transfer rate in a column with a gas–liquid interface was dependent on the number of NPs added in the volume fraction, as well as on the size of the NPs. In the liquid–liquid extraction process experiment conducted by Saien et al., a single drop of Fe_3_O_4_ NP-based FNs was used in a toluene/acetic acid/water system [41]. They achieved 157% mass transfer enhancement at a 0.002 wt.% concentration of Fe_3_O_4_ NPs. Similarly, Wu et al. studied the addition of Fe_3_O_4_ NP-based FNs in ammonia/water bubble absorption under an applied magnetic field [42]. The results showed that the absorption was improved by the combined effect of the FN amount and the intensity of the magnetic field, rather than an individual effect. The effective absorption ratio achieved a maximum of 1.0812 ± 0.0001 at 0.10% mass concentration of Fe_3_O_4_ NPs with 280 mT intensity of the magnetic field, while the mass concentration for ammonia was 20%.

Moreover, Saien et al. studied liquid–liquid extraction using a single drop of Fe_3_O_4_ NP-based FNs under an oscillating field [43]. The ferrofluid concentration was used in the range of 0.001–0.005 wt.%, and the intensity of the magnetic field was in the range of 0.36–1.45 T. An average enhancement in mass transfer was found to be 65%, and a maximum of 121% (for the smallest used drops) was achieved when compared in the absence of a magnetic field. The whole experiment setup (Figure 2A) was designed using a Pyrex glass column (11.4 cm diameter and 51 cm height) as the contactor. At the bottom of this column, a glass nozzle was used to produce different drop sizes. With the help of adjustable syringe pumps, the toluene–FN phase was transferred via a glass syringe and rigid tube into the glass nozzle, while the aqueous phase was filled into the column. In the middle of the drop path along the column, two identical coils (10 cm in length) were placed next to the column. The positions of the coils were varied at different heights of 15, 25, and 35 cm for the generation of drops from nozzle 1, with an FN concentration of 0.004 wt.% and a magnetic field intensity of 0.73 T. All experiments were performed at an ambient temperature of 20 °C (±2), and the flow of mass transfer was from a dispersed phase to continuous phase. The overall mass transfer coefficient (k*_od_*) variation with magnetic field intensity with NP concentration is shown in Figure 2B. The experimental values of mass transfer were obtained in the range of 33–154 (±1.3) μm·s^−1^. The motion of the NPs in the drops without and with a magnetic field is shown in Figure 2C,D, respectively. The best overall mass transfer coefficient was found to be 128 (±1.3) μm·s^−1^ at 25 cm height of the coil, whereas, at 15 and 35 cm height of the coil, the overall mass transfer coefficients were obtained as 118 and 135 μm·s^−1^, respectively.

Furthermore, strategies for mass transfer intensification of a liquid–liquid extraction using a single drop were also investigated by Saien et al. [44]. Similarly, Hoseini et al. studied the mass transfer intensification process for a gas–liquid phase [45]. They investigated the effect of an electric field on the absorption of CO_2_ gas in Fe_3_O_4_ NP-based FNs. A volume concentration of FN less than 0.03% was used in the presence of electric field intensities of 133, 200, and 266 kV/m. The experimental results showed that the presence of the electric field significantly enhanced the mass transfer rate for low-concentration ferrofluid, whereas, for the high concentration of FN, the results were found to be deteriorated. The experimental setup (Figure 3A) used in this experiment consisted of a wooden frame for holding two tubes vertically with different diameters of 6 and 14 mm and a length of 50 cm. Two copper electrodes (28 cm in height and 6 cm in width) were placed for the exertion of an electric field. For measuring the voltage break between the two electrodes, a multimeter was used with an uncertainty of ±0.5 kV. A high-voltage power supply was used to impose an electric field. All experiments were performed at room temperature and one atmospheric pressure. Three different voltages of 10, 15, and 20 kV were provided during the experiment. The syringe was used for injecting a gas (5 mL) into the test tube through the entrance section. As soon as the gas moved upward into the tube, a liquid falling film was formed on the tube wall. The gas was absorbed upon further movement of gas into the liquid, and the length of the bubble was reduced. The cameras mounted on the entrance and exit sections took pictures of the length of the bubble, as shown in Figure 3B. The experiment was conducted by filling the test tube with the base fluid and FN, in the presence or absence of an electric field, and repeated three times. These results showed a positive effect of the electric field on the mass transfer phenomenon for a dilute concentration of FN.

Recently, Maiorov et al. analyzed the impact of the magnetic field gradient on the gas–liquid interface by sparging to permeate hydrocarbon-based FNs [46]. The gas–liquid contact volume was controlled effectively using a magnetic field by foaming the liquid during the sparging of gases and suppressing it using magnetic forces without reducing the mass transfer efficiency. In a typical experiment, Fe_3_O_4_ NPs were prepared and dispersed in water for the preparation of FNs. The aggregates revealed the drying of the FN sample according to TEM analysis (Figure 4A). The dispersion is clearly shown in the image, along with roughly spherical NPs. Figure 4B shows the magnetization of the FN, highlighting the magnetic nature of the sample. Figure 4C shows a schematic view of the experimental setup of the mass transfer phenomenon, constituting various components such as a peristaltic pump, dehumidifier, test tube for FN mass transfer, permanent magnet arrangement, and IR spectrometer. Figure 4D shows the pattern of the magnetic field in the central cross-sectional area over the permanent magnet unit (4 in Figure 4C) swept by a Hall effect sensor. The size of the permanent magnets was 20 mm × 15 mm. The dashed line marks the position of the test tube containing the FN sample. Figure 4E shows the IR spectrum of the sample after the run, exhibiting two main curves at 3000 and 1500 cm^−1^ due to C–H bending and C–H stretching vibrations for alkanes. The test was conducted for 24 min (a) and 48 min (b). The inset shows a detailed view of the respective curve, attributed to hexane. Figure 4F shows the kinetics of the gaseous solute concentration (hexane) in the effluent estimated using an IR spectrometer in terms of the volume of gas transported (a) in the absence of a magnetic field and (b) in the presence of a permanent magnet unit. The indicators demonstrate the locations of the spectral peaks in Figure 4E, with an exponentially fitted black line through the concentration degeneration area. Figure 4G presents the morphology of the gas–liquid boundary throughout the mass transfer experiment (a) in the absence of an external magnetic field and (b) in the presence of permanent magnet unit, demonstrating total removal of foaming due to the magnetic effect of NPs. FNs have already undergone several developments for mass transfer; for example, they are effective in the purification of air, separation of gases from mixtures of gases, gas recovery, and gas transportation [47].

On the other hand, chemical engineering is also concerned with desalination technology. Due to the high cost, nonrenewable nature, and resulting environmental hazards of current methods, it is important to find an alternative water purification method that is inexpensive and environmentally friendly [48,49]. As a result, the development of an intriguing desalination technology is considered a major challenge for researchers around the world [50,51]. Several desalination methods have been developed over the years. The most common are electrodialysis, distillation, and reverse osmosis [52,53]. However, there are some problems with these methods, such as high maintenance costs and secondary chemical waste [54,55]. Several properties such as high absorption, catalytic activity, and reactivity are associated with nanomaterials [56,57,58,59,60,61]. The decomposition of many pollutants into a variety of biological and chemical species, including bacteria, novel pollutants, organic pollutants, and inorganic ions, has been reported worldwide [62,63]. Hence, IONs are a promising tool that can be applied to various wastewater ecosystems, such as carbon nanotubes, zero-value NPs, metal oxide NPs, and nanocomposites [64,65].

## 3. Electrical and Electronics Applications

The explicit control over optical properties, shape, and structure of ferrofluids using external magnetic fields makes them suitable for numerous sensing applications. The use of FNs boosts the motion sensitivity of sensors such as flow meters, accelerometers, vibration sensors, inclinometers, tilt sensors, level and pressure sensors, and various switches [66]. The incident power is one of the important factors affecting the sensitivity of optical transmission based ferrofluidic sensors. In recent years, researchers have studied various ION-based magnetic nanofluid sensors, such as magnetic field sensors, temperature sensors, and sensors for on-chip devices.

### 3.1. FN-Based Magnetic Field Sensors

Due to their flow behavior and high sensitivity, even under a weak external magnetic field, FNs have attracted considerable interest. Variations in the magnetically tunable refractive index lead to two distinctive processes: birefringence (due to the fine structural modifications in ferrofluid under a weak magnetic field) and transverse magneto-optical effect (due to the saturation magnetization of the ferrofluid owing to a stronger magnetic field) [56]. They possess numerous magneto-optic properties such as the birefringence effect [57,58], adjustable refractive index [59], Faraday effect [67], and thermal lens effect [60]. The concentration of Fe_3_O_4_ NPs in the FN affects its optical properties. Some of these properties of FNs have been utilized in magneto-optic sensors to calculate the magnetic field intensity with exceptional sensitivity [61,62]. Magnetic fluid sensors based on optical transmission properties have the fastest-growing application prospects including military, power generation, healthcare, archeological surveying, and aerospace [63]. Generally, magnetic field sensors estimate the magnetic flux intensity of a medium in Tesla or Gauss units [64,65]. For example, Zu et al. recommended a fiber magnetic field sensor built utilizing Fe_3_O_4_ NP-based FN films in a Sagnac interferometer, which has possible uses in switches and modulators [68]. Similarly, Ji et al. proposed V-shaped groove and capillary tube-based sensing systems filled with commercial water-based FNs with discrete Fe_3_O_4_ NPs (~10 nm diameter) to calculate the magnetic field intensity [69,70]. The schematic illustration of the V-shaped groove is shown in Figure 5A. On the other hand, Chen et al. proposed an optical fiber magnetic field sensor, as shown in Figure 5B, where discrete Fe_3_O_4_ NP-based FNs and a single-mode–multimode–single-mode (SMS) design were used [71]. Similarly, Candiani et al. employed a single-mode fiber dipped in a Fe_3_O_4_ NP-based FN to estimate the magnetic field, which was based on the backscattering intensity of the FN in varying magnetic fields [72]. Figure 5C illustrates the system flow diagram used in magnetic field sensing. The cladding holes of a photonic crystal fiber (PCF) filled with FNs was employed to estimate the magnetic field intensity. It utilized variations in the refractive index and birefringence to detect the magnetic field with high resolution [73]. Moreover, Homa et al. proposed a simple and economical fiber-optic magnetic sensor built using an extrinsic Fabry–Perot interferometer (EFPI) infused with a Fe_3_O_4_ NP-based FN that could easily sense a magnetic flux density of 0.5–12mT magnitude [74]. Figure 5D shows a schematic of the assembled extrinsic Fabry–Perot interferometer (EFPI) sensor filled with Fe_3_O_4_ NP-based FNs.

Furthermore, Huang et al. designed a novel magnetic sensor employing a thin-core fiber (TCF) interferometer enclosed by ION-based FNs, as shown in Figure 6A [75]. The modal interferometer was formed by intertwining a TCF of a particular length between two single-mode fibers in order to control the resonant wavelength or interference condition. The two TCF interferometers connected in the fiber had differing dip wavelengths, indicating the multiplexing capability of the sensor. The author found the transmission dip wavelength of the interferometer to be shifted, achieving a highest sensitivity of 11.8 pm/Oe. Similarly, Dudus et al. designed an optical fiber magnetic field sensor using ION-based FNs (as shown in Figure 6B) [76]. Unlike other such devices that are based on the magneto-optic effect, this novel sensor was based on the physical movement of an ION-based ferro-nanofluidic shutter due to the application of the magnetic field. When a magnetic field is applied to a ferro-nanofluidic shutter, it starts to move, which varies the amount of light transmitting across the magnetic sensor; therefore, it measures the changes in the intensity of light across the sensor due to the applied magnetic field. The advantage of this sensor is that it is not affected by interferometric noise, which is a concern for other fiber sensors. When a magnetic field is applied to the proposed extrinsic fiber-optic sensor, the casually orientated IONs of the FN are aligned and combine their individual magnetic moments, forming a ferro-nanofluidic “shutter” that moves toward the region of higher magnetic flux between two single-mode optical fibers (SMFs). The optical ferro-nanofluidic shutter comprises a plug of diluted and opaque ferrofluid directly touching a plug of visibly clear and immiscible fluid of glycerin and water. The ferrofluid (CHE006, Rapid Electronics Ltd., Colchester, UK) was diluted by mixing with mineral oil in a 1:30 ratio. The dilution ratio was chosen such that the FN did not leave any residue on the wall of the glass channel, in addition to being opaque enough to attain high optical attenuation. The refractive index of ferrofluid and glycerin fluid was matched at 1.450 through their careful dilution. This minimized the undesired interferometric effects, thus eliminating interferometric noise in the sensor. The ferro-nanofluidic shutter system was actuated in a fluidic channel using another liquid plug consisting of 100% pure ferrofluid (CHE006, Rapid Electronics Ltd., Colchester, UK). The third ferrofluid plug was immiscible with the glycerin/water solution, and it moved due to the applied magnetic field. The investigation revealed that the sensor operates in the range of 0 to 10 mT with an optical attenuation of 28 dB at 10 mT. This proposed magnetic sensor offers a simpler and more economical setup for an optical magnetic sensor. This sensor can be further miniaturized and made more cost-effective by using broadband LED and a cheaper detector.

On the other hand, Mitu et al. proposed a novel magnetic field sensor using Fe_3_O_4_ NP-based FNs for detecting magnetic field strength in the range of 100 Oe to 160 Oe [77]. Figure 6C shows the cross-sectional view of the novel magnetic field sensor using air holes and silica cladding with Fe_3_O_4_ NP-based FNs. The proposed magnetic FN sensor was fabricated using a sol–gel technique with a dual circular elliptical core cladding photonic crystal fiber (EC-PCF) structure; the major and minor axes were 0.7 µm and 0.5 µm, respectively, and the base was built using silica glass (Figure 6C). The optimal values of different parameters were chosen by tuning the proposed model. The distance between the centers of two adjacent air holes, called the lattice pitch (ᴧ), was chosen as 2 µm, and the applied wavelength range was in the range of 1400–1700 nm, as illustrated in Figure 6C. Flexibility in the design was achieved through the injection of air holes within the PCF or silica glass. The outer circular layer was the perfectly matched layer (PML) of the sensor structure with the radius being 10% of the cladding region. The dual core inside the sensor structure was filled with magnetic FNs on the same axis as the air holes. The use of ION-based FNs helped to achieve a high sensitivity of 83,268.46 nm/RIU and 83,188.25 nm/RIU for X and Y polarizations, respectively, at a room temperature of 20 °C and concentration of 1.2 g/mL. The sensor was found to possess a simple structure, low cost, and high accuracy, being useful for microfluidics technology and optical fiber technology, which opens a path to a wide range of applications such as telecommunication, sensing and optical communication, and microfluidic devices.

Similarly, Luo et al. proposed a magnetic sensor using a magnetic fiber coupler (MFC) surrounded by a magnetic fluid [78]. A schematic configuration of the proposed magnetic field sensor is shown in Figure 7A. Both ends of the capillary were sealed with UV adhesive to prevent leakage or evaporation of the magnetic liquid. In this work, a water-based magnetic nanofluid with a density of 1.18 g/cm^3^ and a saturation magnetization of 200 Oe provided by Beijing Sunrise Ferrofluid Technological Co., Ltd., Beijing, China, was used. Light from a supercontinuum broadband source (SBS, Wuhan Yangtze Soton Laser Co., Ltd., Wuhan, China) was coupled into P1, and the output light from P4 was collected using an optical spectrum analyzer (AQ6370C). The sensor structure was placed between two poles of an electromagnet that created a uniform magnetic field with a non uniformity of less than 0.1% within the sample area. The strength of the magnetic field was adjusted by adjusting the size of the supply current. The direction of the magnetic field was perpendicular to the fiber-optic axis. A tesla meter was used to measure the strength of the magnetic field, and the ambient temperature was kept at 18 °C. The MFC was extremely sensitive to the surrounding RI due to the large evanescent field outside the fiber surface in the fused area. Since the RI was sensitive to magnetic fields, the MFC surrounded by MF could be used as a magnetic field sensor. Additionally, this also implied the possible applications of the proposed structure to optical power dividers, optical filters, and other optical devices. Moreover, Li et al. studied the effect of incident power in order to investigate the magnetic field sensitivity using a magnetic field sensor [79]. When an incident power of different intensity was transmitted through the magnetic fluid sensor, a variation in the sensitivity of the magnetic field was obtained and divided into four stages: first decreasing sharply, then increasing, before gradually decreasing, and finally tending toward a small stable value. This was due to structural changes in the magnetic NPs of the FN, as well as due to the Soret effect and Hall effect. The change in sensitivity can be varied by applying different incident powers. For example, when a high sensitivity is required in a weak magnetic field, a large incident power can be selected, whereas, in a strong magnetic field, high sensitivity is achieved by selecting a small incident power.

Furthermore, Samian et al. measured the magnetic field strength using magnetic fluid and fiber bundles by recording the changes in the refractive index of the magnetic fluid [80]. In this work, a magnetic liquid with a concentration of 2.5% was used. With the proposed sensor, a magnetic field in the range of 25.3 mT to 83.5 mT was successfully recorded with a resolution of 3.3 mT. Recently, Nandy et al. used magnetic nanoemulsions (MNEs) to study the defect sensitivity in carbon steel under different functional entities such as anionic surfactants, copolymers, and a weak polymer electrolyte [81]. The defects on carbon steel samples were artificially created (rectangular slots). The MNEs formed linear chain-like structures, which were visualized using atomic force microscopy and phase-contrast optical microscopy. The variations obtained in the defect sensitivity were made accessible by color changes in the defect region under different stabilizing entities. Figure 7B shows the experimental setup of magnetic flux leakage (MFL), where samples were magnetized with the help of an electromagnetic yoke (M/s Magnaflux). To reduce spurious reflections and to provide a constant background intensity, a white sheet was placed on the surface of the carbon steel samples. A thin cuvette was filled with the MNEs, and the internal spacing was 1 mm. To simplify manufacturing, as well as optimize liquid volume, color contrast, and defect width estimation accuracy, the optical path length of the MNE-based thin-film sensor was fixed at 1 mm to minimize magnetic field variations due to increased effective liftoff. The generated optical patterns were captured using a Nikon D700 digital camera and later analyzed using ImageJ software. Hall probe measurements (Model 5080, M/s F.W. Bell) were also conducted at a fixed liftoff of 1 mm to measure the normal components of the MFL.

### 3.2. FN-Based Temperature Sensors

Since temperature sensors have significant applications in the field of science and innovation, several researchers have been investigated them. With the improvement of assembly methods, temperature sensors can now be designed utilizing FNs. For example, Zhang et al. designed a temperature sensor using an FN thin film based on optical transmission [82]. The magnetic field was applied in the parallel plane to the FN thin film; magnetic chains were formed in the direction of the magnetic field to cause optical transmission suppression. The change in transmission of the FN was due to a change in ambient temperature, which could be used to construct the temperature sensor by measuring the transmission power of the laser. The temperature sensor arrangement is shown in Figure 8A. In the experimental setup, an FN thin film was placed between the two solenoids, which applied a magnetic field in the parallel direction to the plane of the FN thin film. The FN thin film acting as a probe was made up of NPs (average size of 10 nm) dispersed in water (6.47% volume concentration) and sealed in a 3.3 × 1.7 cm glass cell with a thickness of 7 μm. The magnetic field strength was measured using a teslameter and adjusted by changing the amplitude of the square-shaped wave. A magnetic field of strength 40.9 mT was used in the experiment. The laser light of 650 nm wavelength and the magnetic field were perpendicular to each other and to the FN thin film. The intensity of transmitted light was detected in the direction of light propagation. The optical signal was detected by a silicon photovoltaic cell and shown using an oscilloscope. The electrical signal of the applied magnetic field was also shown using the oscilloscope connected to the teslameter. The ambient temperature around the FN thin film was changed using an electric heater and detected using a thermocouple. The temperature sensor could be evaluated by investigating the change in optical transmission with the developing temperature. After the examination of its affectability, it was realized that this temperature sensor structure was reasonable for high-temperature usage (above 60 °C).

Similarly, Zhao et al. proposed a concurrent technique for constructing a dual-parameter sensor using water-based Fe_3_O_4_ NP-based FNs [83]. The sensor can be used for determining both magnetic field and temperature using a photonic crystal fiber Bragg grating (PCFBG) with two-resonant peaks, filled with Fe_3_O_4_ NP-based FNs, as shown in Figure 8B. Further development of temperature sensors was demonstrated by Pathak et al. using Fe_3_O_4_ NP-based FNs [84]. A high-precision 3.7 mK temperature sensor was developed using FN bearings. The device was based on the expansion/contraction of air as a working fluid. The coefficient of friction of FN bearings was very low (µf = 0.002), which provided a perfect seal and frictionless moving. The sensor can be used for applications requiring high-accuracy temperature measurements, with a high sensitivity of 3.7 (± 0.2) mK and easy customization. For trapping the air, a glass bulb with a temperature-sensitive base of 115 mL was used. A capillary of 30 cm length and 5 mm diameter was fused on top of the glass bulb for the passage of air expansion. For the smooth movement of air into the bulb without interaction with the magnet, the bottom part of the capillary was slightly converged.

On the other hand, FN-based temperature sensors also have applications in biomedical engineering such as the treatment or diagnosis of cancer and MRI. Accurate temperature measurements are necessary for safe and effective thermal treatment for cancer and other related diseases. The Brownian relaxation time of NPs is one of the methods to measure the temperature. For example, Perreard et al. demonstrated a new noninvasive method to measure the temperature of MNPs in the microenvironment due to the Brownian relaxation time of NPs [85]. The MNPs under the applied magnetic field at various frequencies induce magnetization, which helps in determining the relaxation time of MNPs. This method was found to be accurate and effective as compared to previous other methods at higher applied field frequencies. Similarly, Draack et al. studied the dynamic characteristics of MNPs, which was found to be temperature-dependent [86]. Both Neel and Brownian relaxation mechanisms affected the dynamic characteristics of MNPs, as the harmonics spectrum was found to be influenced by the change in temperature or viscosity. Temperature-dependent magnetic particle spectroscopy (MPS) can be used for magnetic particle imaging (MPI) and magnetic hyperthermia because the MPS data determine the dominating relaxation mechanism in MNPs. Moreover, Wu et al. reviewed the studies related to the temperature-dependent harmonic spectra in MNPs [87], where MPS data were taken into account to explain the magnetic relaxation mechanism and explore its application for a bioassay testing kit as a cheap, portable, and highly sensitive device.

### 3.3. FN-Based Tilt Sensors/Inclinometers

Numerous research articles discussed the development in the field of inclinometers and tilt sensors. These sensors are designed and engineered according to the concept of the gravitational force to measure the tilt angle. The most common problem in these sensors is their bulkiness, making their use challenging in several circumstances. FNs can be used to address this problem so as to design compact, accurate, and cost-effective tilt sensors and inclinometers. Ando et al. proposed an inertial sensor and actuator employing Fe_3_O_4_ NP-based FNs, which used the fine spikes and the corresponding surface perturbations around equilibrium conditions in FNs due to the Rosensweig instability in vibration sensors [88,89]. Furthermore, the shift in equilibrium configurations due to nonmagnetic fluid flow in a magnetic nanofluid spike could be applied for quantifying the flow in pipes [90]. Ando et al. also demonstrated a seismic sensor that combined an ionic polymer–metal composite (IPMC) dipped in a flask filled with Fe_3_O_4_ NP-based FNs and encircled with four symmetrically positioned permanent magnets [91]. The response of the IPMC membrane was a function of both the viscosity and the density of the FN. The fundamental reason for using a magnetic nanofluid was its inherent ability to change its shape and density as a function of an externally applied magnetic field. Similarly, Li et al. proposed an all-fiber optical current sensor (AFOCS) system based on FNs and a single-mode–multimode–single-mode (SMS) design having a no-core fiber (NCF) [92]. The proposed sensor consisted of a piece of NCF (length ~7.8 cm; diameter of 61.5 μm) intertwined between two pieces of SMF, which was then placed into the center of a capillary tube (10cmlong; inner diameter of 0.3 mm) filled with ION-based FNs by capillary force. The capillary tube was sealed on both sides using UV glue. The current sensitivity of the demonstration sensor was found to be 2.120 dB/A, with linearity from 2.5 A to 6.5 A and a detection limit of 200 mA.

On the other hand, DeGraff et al. proposed a novel tilt sensor using a transformer setup with a magnetic nanofluid as its core, as shown in Figure 9A [93]. The FN moves within the sensing and exciting coils of the transformer, owing to the force generated through the tilting of the swinging FN mass. When tilted, the mass draws the adjustable tube through rollers, driving the FN to the sensing coil side, which induces a voltage across it. Initially, the FN mostly settles in the exiting coil if the sensor is at rest without any tilt, as shown in Figure 9B. Upon tilting, the FN mass starts to shift, causing the flexible tube to be drawn by the rollers, which squeezes the FN into the sensing coil. This excites the FN in the sensing coil and causes a voltage across it. The value of the induced voltage is dependent on the length of the FN present in the sensing coil. The operation of the sensor is based on overcoming the attractive force between the exciting coil and the FN, the gravitational force and the friction between the tubes, and the squeezing of the FN.

Similarly, Martin et al. developed and examined a force sensor using an FN core-based transformer [94]. The proposed force sensor had a Teflon reservoir whose open top was covered with a silicone membrane. Figure 9C,D illustrate the design and fabrication of the proposed force sensor. The primary and the secondary coils were spiraled across the displacement tube connected to the reservoir. The FN in the displacement tube acted as the transformer core. Whenever force was applied on the silicone membrane, it caused the magnetic nanofluid present in the Teflon reservoir to flow across the primary coil toward the secondary coil, which produced a voltage in the secondary coil depending on the FN level in the tube. The sensor was based on a nonlinear relationship between the force exerted on the silicone membrane and the voltage produced in the secondary coil during the loading and unloading of the sensor. The authors tested the proposed force sensor at three primary frequencies of 60, 100, and 120 kHz, and the sensor effective range was found to be 0.1–2.5 N for an output voltage of 493–657 mV_pp_. The sensitivity of the proposed force sensor was found to have an average value of 68.3 mV/N across its entire range at an excitation frequency of 60 kHz. This was the first force sensor reported to employ a transformer with FN cores.

### 3.4. MEMS and On-Chip Components Using FN (FN Based On-Chip Transformers)

Lab-on-a-chip devices are essentially a subcategory of microelectromechanical system (MEMS) devices which integrate mechanical elements, sensors, actuators, etc. on a single substrate, sometimes called “micro-total analysis systems” (µTAS) [95,96,97,98,99,100,101,102,103,104,105]. The use of FNs in lab-on-a-chip devices offers excellent improvement in their characteristics such as sensitivity and biolabeling capabilities as a function of their excellent optical, magnetic, heat transfer, and electrical properties. An on-chip transformer consisting of a ferrofluidic magnetic core and a solenoid coil was reported by Tsai et al. [106]. Figure 10A illustrates the transformer on a capillary filled with oil-Fe_3_O_4_ NP-based FNs. The solenoid-type coils were assembled by MEMS technology, whereas the oil-Fe_3_O_4_ NP-based FNs used in the magnetic core were synthesized using the chemical co-precipitation technique. The performance metrics of an MEMS transformer with a ferrofluidic magnetic core were simulated and estimated in the frequency range of 100 kHz to 100 MHz. Figure 10B presents the inductances of the magnetic coils of the on-chip transformers with several types of magnetic cores (air and Fe_3_O_4_ ferrofluids of 0.25, 0.5, 0.75, and 1 M). The inductance dropped swiftly in the frequency range of 100 kHz to 15 MHz because of the skin effect of the coils, whereas, for 15–100 MHz, it steadily increased and approached a maximum value at the resonant frequency. It was also found that the inductance of the magnetic core increased linearly with the increase inFe_3_O_4_ NP concentration. Figure 10C,D present the simulated and measured outcomes of the coupling coefficients of the on-chip transformers with varying magnetic cores. With the increase in the concentration of Fe_3_O_4_ NPs, the coupling coefficient also increased. The coupling coefficient increased quickly to 5 MHz and then increased steadily afterward. Hence, the use of ION-based FNs in the core of an on-chip transformer strengthens the inductance of the magnetic coil and the coupling coefficient of the transformer.

Moreover, Lazarus et al. created magnetic components with high-quality factors such as magnetic core transformers (nine-turn solenoid), inductors (12-turn toroid), and wireless power coils (four-turn planar coil) using FNs, as shown in Figure 11A–C, respectively [107]. They employed a novel multistage fabrication technique based on 3D printing accompanied by fillings with FNs. The first step was to print the multilayer microfluidic channels employing stereo–lithography. Such devices comprised two conductive layers with vertical vias. The conductive layers were 2.5 mm apart, the liquid metal channels were covered with 1 mm of polymer above and below, and the cross-section of each channel was 1.5 mm. Next, the microfluidic systems were filled with liquid gallium alloy, eutectic gallium–indium (EGaIn), and FNs at room temperature to fabricate transformers, inductors, and wireless power coils. The investigation showed that the use of magnetic nanofluid increased the inductance density three fold, in addition to coupling gains for transformers and wireless power coils. This was the first system of its kind fabricated using the 3D printing technique [107].

Similarly, Keneth et al. fabricated 3D-printed, ferrofluidic, soft robotic actuators with complex shapes and remote actuation under an applied magnetic field [108]. The proposed actuator consisted of a 3D-printed elastic polymer tube filled with an oil-based FN such that the FN was movable to different locations in the actuator, thus affecting the actuator response. They demonstrated two novel actuators: (i) a hand-like actuator, and (ii) a worm-like actuator. The hand-like soft actuator could carry out different finger movements through the spread of magnetic nanofluid from a reservoir in the palm. The hand-like actuator demonstrated the advantage of using hollow channels in regulating the movements. The inch-sized worm-like robotic actuator consisted of a bent tube filled with FNs connected with 3D-printed footpads, which functioned as directional adhesives. The two footpads connected to each worm-like actuator comprised materials with dissimilar friction coefficients. The study reported the effect of both the ferrofluidic flow and the mechanical characteristics of the elastic polymer on the performance of proposed actuators, demonstrating complex motions akin to human hands and worms. This study can act as a starting point for further research on ferrofluid-based soft robotics actuators with potential for future improvement through the use of electromagnets, valves, movement sensors, etc.

Furthermore, Doganay et al. demonstrated a unique FN-based revolving permanent magnetic actuator (PMA) system for pumping of different chemicals into micro-sized semicircular channels [109]. The actuator comprised two sets of permanent magnets, a rotor-shaped structure, and a microchannel with a square cross-section placed on the actuator. Fe_3_O_4_–water FNs were used as the working fluid inside the microchannel. The investigation showed that, by varying actuator speed (7.5–30 rpm) and ION concentration (1–3% by volume), FNs inside the microchannel could be operated with speeds of 58.7–940 µm/s, corresponding with flow rates of 0.56–9.02 µL/min. The electrical power consumed by the proposed actuator was found to be constant. For the performance test at 15 rpm, flow rates could be increased up to threefold by only improving the NP concentration of the magnetic nanofluid from 1% to 3% by volume, resulting in a 1.4-fold increase in the viscosity. The test results established that the novel actuator system was suitable for pumping fluids in various microfluidic applications such as cell sorting devices (6.1 µL/min), low-flow drug delivery systems (1–10 µL/min), and pathogen detection systems (3–5.83 µL/min).

### 3.5. Cooling for Electronic Devices and Photovoltaic Thermal (PVT) Systems

FNs are capable of heat transfer; therefore, they are used as coolants in electronic devices and pipes such as photovoltaic thermal (PVT) systems, LEDs, and pulsating heat pipes [110,111,112,113,114]. Similarly, Fe_3_O_4_ NP-based FNs also seem to be a very good option as they have been found to be excellent coolants in PVT systems, LEDs, and pipes [115,116,117]. Ghadiri et al. investigated a Fe_3_O_4_ NP/water FN system for the cooling of a PVT system to enhance its efficiency under constant and alternating magnetic fields [115]. A ~76% improvement in the overall efficiency of the PVT system was achieved using a 3 wt.% concentration as compared to water as a coolant. Similarly, Heidari et al. investigated the effect of Fe_3_O_4_ NP-based FNs on the power enhancement, surface temperature, and electrical efficiency of the PV cell [116]. Figure 12A shows a schematic diagram of the experimental setup, where metal halide lamps were used for the light source, mounted at a height of 50 cm from the surface of the PV cell. The temperature of the cell surface was recorded every 10 s using a thermometer during the experiment. Two permanent magnets were attached to the blade for the induction of the magnetic field. Figure 12B shows that the temperature increased with time in all cases. However, when water was used as the cooling medium, the heating of the PV cell surface decreased, while, when FN was used as the cooling medium, the temperature was decreased further. Moreover, as the concentration of the FN increased, the surface temperature decreased. Under a rotating magnetic field (RMF), the surface temperature decreased with the increase in the concentration of FN (Figure 12C). Overall, FNs under an RMF showed a tremendous improvement in the cooling of PV cells as compared to water and FN in the absence of an RMF. The improvement in the thermal efficiency (ɳ) was found to be 17.8–30% when 0.05% (*w*/*v*) FNs were used under an RMF. The maximum power generated was found to be 3.5 W, and the improvement in power was found to be 47.5% at 880 mT.

Moreover, Seo et al. designed a high-power LED cooling system using FNs and experimentally investigated the heat transfer and illuminance performances of this system for vehicle headlamps [117]. The thermal resistance characteristics were investigated at an input voltage of 18.9 V. The total thermal resistance with the FN after 3600 s was found to be decreased by 15.5% and 18.2% as compared to air and water, respectively. Additionally, the thermal resistance between the surface and the junction of the high-power LED cooling system with the FN was found to be decreased by 44.8% and 29.3% as compared to air and water, respectively. On the other hand, the illuminance of the high-power LED cooling system with the FN after 3600 s increased by 67.5% and 66.1% as compared with air and water, respectively.

## 4. Environmental Engineering Applications

Various industries produce effluents in tons that directly flow into the natural water sources and, thus, pose a great danger to the environment. Some natural phenomena and human activities also pose threats to the natural sources of water. The increase in the direct disposal of these effluents into water raises the risk to human, animal, and plant health in many ways. Therefore, wastewater treatment has attracted great attention because of the increase in water pollution through many avenues such as from chemical industries [118,119,120,121,122,123]. There are various methods for water treatment such as settling, filtration, disinfection, and coagulation [124,125,126,127]. Among all methods, coagulation is an important process for water and wastewater treatment. The suspended particles are electrically charged molecules that can be neutralized by coagulants and become unstable, eventually settling down due to gravity [126,127]. Different coagulants are used in water treatment processes such as inorganic metal salts, synthetic organic polymers, natural coagulants, magnetic NPs and their composites, clays, chitosan biopolymer, and tannin-based coagulants [128,129,130,131,132,133,134,135]. Alums are also extensively used in the treatment of water because of their availability, low cost, and efficient elimination of turbidity. However, their ecotoxicological impact after treatment is also a big concern [136].

Hatamei et al. introduced ION-based FNs for an effective water treatment process, which was not only useful in turbidity removal but could also remove heavy metals ions from the water and wastewater [136]. Maghemite NPs were synthesized by co–precipitation; then, coagulant studies were conducted, and the effects of NP concentration, temperature, and reaction pH on coagulation performance were studied. The performance of maghemite NPs was also evaluated in relation to the removal of heavy-metal ions and bacteria. A reusability test showed that the MNPs could be recovered after the water treatment process. After treatment, these MNPs could be effectively removed from water samples by applying an external magnetic field. Figure 13A shows the removal percentage of heavy-metal cations at different concentrations (1 μg·mL^−1^ and 4 μg·mL^−1^) of heavy-metal ions (Cd^2+^, Co^2+^, Mn^2+^, Fe^2+^, Pb^2+^, Zn^2+^, Ni^2+^, and Cu^2+^) following their addition to a real water sample. The removal mechanism of the ions was based on sorption [137], cation exchange [137,138], and hydroxide formation [139] on the surface of MNPs. The study was carried out using 0.15 g of MNPs in 200 mL of the water sample at 25 ℃. The initial and final concentrations of water sample were analyzed using anatomic absorption spectrophotometer (AAS). The percentage removal was calculated using the following equation:(1)$$\mathrm{Removal}\;\left(\%\right)=\frac{C_{i}-{\;C}_{f}}{C_{i}}\times 100$$
where C_i_ is the initial concentration, and C_f_ is the final concentration.

Figure 13B shows the percentage removal of phosphate ions from 100 mL of three different water samples, with pH maintained at 7.2, and a nano adsorbent concentration of 0.15 g. The findings showed that 72–80% of phosphate ions were removed from all samples. The presence of bacteria in the water leads to pollution, necessitating their removal. Anantibacterial study was conducted against *coliformsbacteria* (CB) and *fecal coliform bacteria* (FCB) using MNPs [136]. The raw samples were taken, and the number of bacteria was calculated using the standard method. The standard test method was repeated after treating the real water samples. The number of bacteria was reduced by more than 98% and 97% in the case of FCB and CB, respectively, indicating the strong antibacterial activity of MNPs. The percentage removal (see piechart) and the number of bacteria before and after the treatment of water samples are shown in Figure 13C.

Similarly, air pollution due to the emission of particulate matter (PM), especially PM_2.5_ from fossil fuel-based energy conversion devices, is very common, which has now become a worldwide environmental concern. The combustion engines and boilers are the most critical when it comes to generating PMs in the exhaust gas emitted due to the combustion of fuel. Recently (in 2020), Kuwahara demonstrated that a magnetic nanofluid (Fe_3_O_4_ NP/water-based FNs) filter combined with nonthermal plasma (NTP) could remove diesel particulates from the exhaust gas emitted from a diesel engine without causing any loss in the pressure [140]. Figure 14A shows the schematic of the designed ferrofluid filter and the image of NTP discharge from the spikes on six lumps of the FN. Simultaneous NTP discharges from the spikes on six lumps of the FN were obtained to facilitate PM removal, as shown in Figure 14B. The size and shape of the spike was observed to be a square bottom, i.e., 2 mm × 2 mm, with a height of 1.5–2.0 mm. The three sets of permanent magnets generated a magnetic flux density of 490 mT on their surfaces. The FN was suspended on the wall of a flow channel using a magnetic force, which acted as an FN filter. The surface area of the FN could be increased as a function of the spiking property of the FN. The PMs were collected on the surface of the FN without any re-entrainment as the exhaust gas containing PMs passed over the FN. Moreover, the PM removal was enhanced by NTP discharge generated on the spikes of the FN. The spikes were observed along the magnetic field lines. The flat surface of the FN started to deform above the critical magnetic flux density. The PMs were collected using inertial and gravitational forces, whereas fine particulates such as NPs were collected using Brownian diffusion. The PM removal efficiencies were found to be 44%, 85%, 99%, and 100% for PMs with diameters d_p_ > 0.3 μm, d_p_ > 0.5 μm, d_p_ > 1.0 μm, and d_p_ > 2.0 μm, respectively.

## 5. Mechanical Engineering or Magneto-Rheological Applications of FNs

FNs are also capable of providing materials with good performance to manufacture seals for machine tools. Generally, magneto-rheological fluids (MRF) are used for sealing because of their high capacity and low friction. The sealing capacity of seals using MRF was found to be more than 45 kPa per ring, while the friction of seals in bearings was found to be 8 N·m. This is considered high for seals used in the precision spindles of any device, as they consume more energy and, hence, generate a large amount of heat [141]. Therefore, MRFs are currently getting replaced by ION-based commercial FNs such as Ferrotec in seals because they reduce the energy consumption [141,142,143]. The category of ferrofluidic seals includes lubricants, dynamic sealing, dynamic process seals, exclusion seals, FN film bearings, braking, and environmental seals for pollution control [144,145,146,147,148,149].

Recently (in 2020), van der Waal et al. demonstrated that the service life of a Ferrotec FN-based rotary seal which dynamically seals pressurized water successfully could be improved and controlled by implementing an FN replenishment system in its design [143]. It was observed that the sealing capacity decreased as the FN in the seal ring degraded over a period of time. This led to seal failure when the sealing capacity decreased with respect to the operational sealing pressure. Moreover, the static sealing capacity increased over time, which was partially attributed to the migration of particles through the carrier liquid toward higher magnetic field intensities. Similarly, Figure 15A shows the practical implementation of an actuator where two hybrid FN seals are used. The FN seals function as an over-pressure valve seal, which is open upon the pressure reaching a limit. The reflux channels and clearance seals are used for flow restrictions. Such hybrid FN seals are suitable for micro-devices where high pressure is required with a low friction force [150].

Furthermore, MRF-based dampers have also been used as a damping medium [151]. However, MRFs have not been successful in damping as they consist of large-sized particles that settle in the medium over time, while their rheological properties also change with time. On the other hand, the magneto-viscous effect in ION-based FNs due to the increase in viscosity in the presence of a magnetic field has been explored in bearings, rotating shafts, brakes, and damping in mechanical systems [150,152,153,154,155]. Pinho et al. evaluated the viscous damping coefficient by varying the volume of FNs in a seal with a steady magnetic field [154]. It was found to be dependent on the local viscosity, shear rate, magnetic field, and frequency. The experimental value (dotted line) of damping was compared with a model (solid line), showing good agreement at a lower volume of FNs (Figure 15B). However, at a higher volume of FNs, a higher value of damping was estimated. In addition to biomedical applications, some important advanced technological applications of ION-based FNs in various fields of engineering are summarized in Table 1.

## 6. Conclusions, Challenges, and Future Perspectives

ION-based FNs possess tremendous potential to be used for advanced technological applications in different engineering fields. IONs, especially Fe_3_O_4_ NP-based FNs, can be used for mass transfer applications in gas–liquid columns, which is a very important process in various chemical engineering systems. On the other hand, the removal of heavy metals from wastewater and surfactant/organic solvents from a mixture can be achieved using IONs. The high adsorption property of magnetic nanocomposites as coagulants prepared using IONs can be exploited for the adsorption of heavy metals in the water treatment process. Similarly, Fe_3_O_4_ NP-based FNs can be used for the treatment of water by decreasing the turbidity and chemical oxygen demand, while also eliminating heavy-metal cations and anions. ION-based FN applications have also been explored in mechanical engineering disciplines such as sealing and damping, where they are used to dampen vibrations in loudspeakers, hard-drive rotary seals, and other rotating shaft motors. Moreover, ION-based FNs have also been recognized as intriguing FNs due to their unusual optical features, such as refractive index tunability in a magnetic field, field-dependent transmission, birefringence effect, dichroism effect, and thermal effect. In addition to improving the performance of inclinometers, accelerometers, and flow meters, they are used to improve the performance of tilt, vibration, pressure, and level sensors, as well as different switches. Similarly, they are also used as coolants for enhancing the performance of electronic devices and PVT systems. Furthermore, researchers are mainly drawn to this exciting field of ION-based FNs as it involves microstructure analysis, intrinsic and field-dependent properties as a function of temperature or magnetic field, and applications in sensing technologies and integrated devices. In terms of ION-based FN applications in sensing technologies and integrated devices, breakthroughs in optical instruments, such as modulators and optical switches, adjustable optical gratings, optical fiber sensors, and coarse wavelength-division multiplexing, are particularly noteworthy. Due to their advantageous qualities, such as high sensitivity, compactness, distant sensing capabilities, and adaptation to hostile conditions, optical fiber sensors based on ION FNs have been widely researched and developed.

Although ION FN-based optical fiber sensors have received substantial attention and exhibit considerable promise, there are still several issues with their viability, repeatability, and stability. Only low-speed applications can be considered due to the poor response time of FNs (more than milliseconds). Additionally, ION FN-based optical fiber sensors still exhibit temperature sensitivity when used for practical measurements, necessitating additional methods or apparatus to account for the temperature cross-sensitivity. Nevertheless, these issues can be successfully resolved by enhancing the characteristics of FNs and implementing improved fiber sensing structures. The performance of ION-based FNs depends on the thermal conductivity, stability, viscosity, and magnetic properties. However, surfactants or other chemicals can be employed to coat the surface of IONs to improve their viscosity and stability. Hence, modified IONs can be expected to exhibit improved properties as compared to their bare counterparts. The colloidal stability of FNs is crucial as it influences many associated properties. The strong magnetic dipole–dipole interactions between particles make it challenging to create stable FNs. Moreover, designing IONs with efficient surface coatings that offer the best performance in different technological applications is a significant challenge. The scaling and safety of large-scale ION manufacturing techniques are additional concerns. However, a one-step method can be used for producing highly stable ION-based FNs, whereas a two-step method can be used for large-scale production.

## Figures and Tables

**Figure 1 molecules-27-07931-f001:**
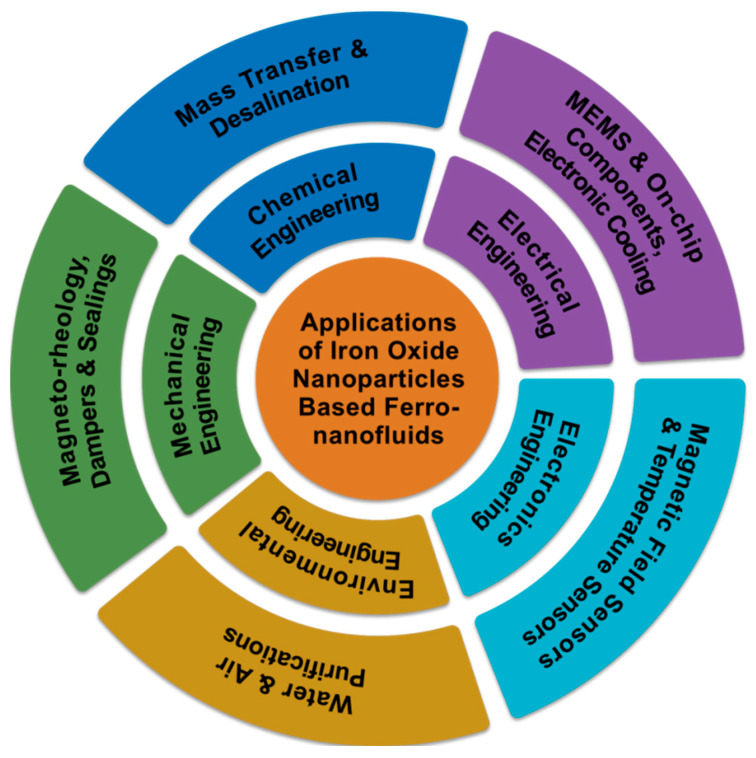
Various advanced technological applications of ION−based FNs concerning different engineering fields.

**Figure 2 molecules-27-07931-f002:**
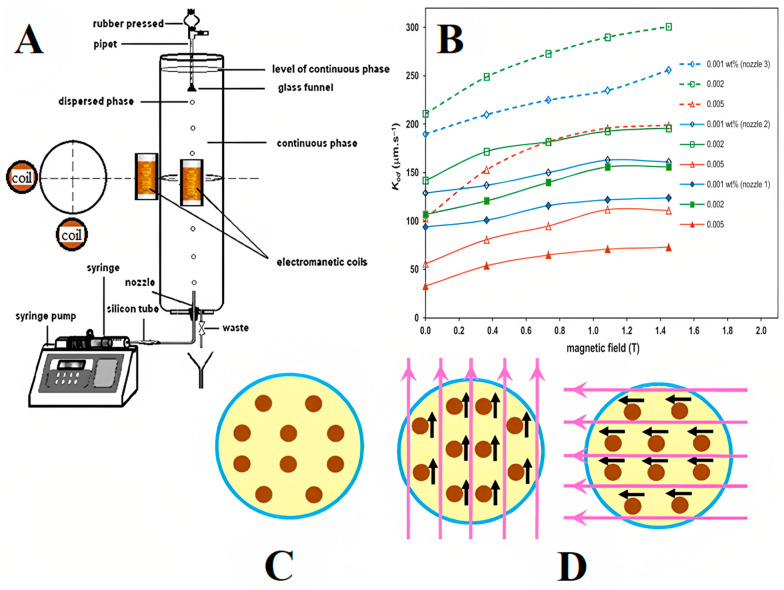
(**A**) Details of experimental setup; (**B**) variation of overall mass transfer coefficient with magnetic field intensity for different nozzles and NP concentrations; NPs in drops (**C**) without magnetic field and (**D**) with applied magnetic field. Reprinted with permission from Ref. [43]. Copyright 2015, Elsevier.

**Figure 3 molecules-27-07931-f003:**
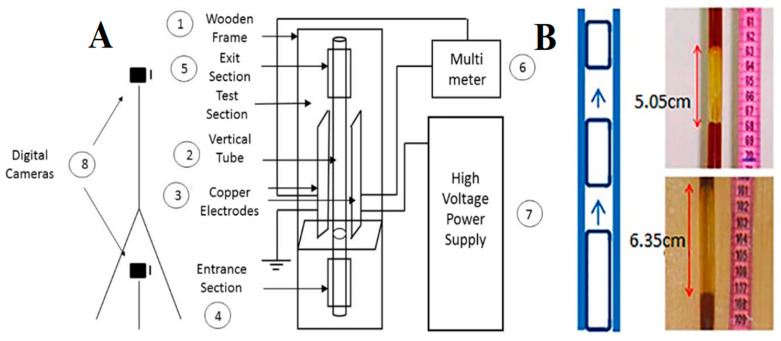
(**A**) Experimental setup; (**B**) digital picture of gas bubble length in the vertical pipe. Reprinted with permission from Ref. [45]. Copyright 2019, SciELO, Brazil.

**Figure 4 molecules-27-07931-f004:**
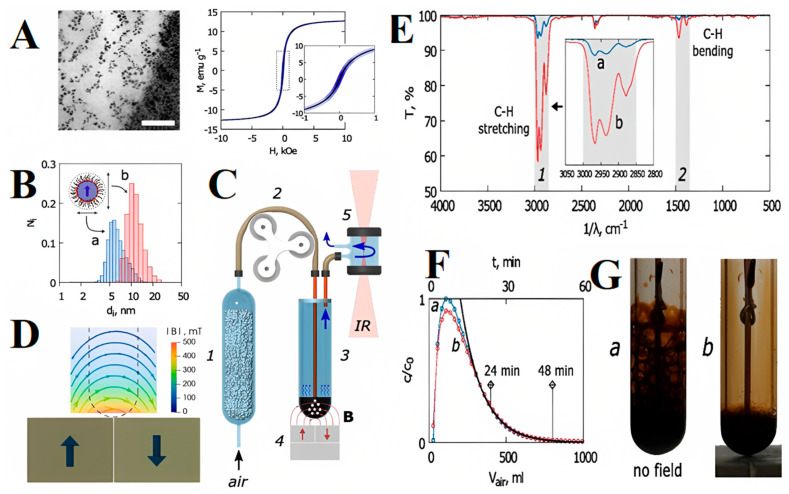
(**A**) Left: TEM image of FN sample (scale bar: 100 nm); right: magnetization curve of the FN sample; (**B**) size distribution (**a**) and hydrodynamic size (**b**) according to DLS analysis of Fe_3_O_4_ core magnetic NPs; schematics of the (**C**) mass transfer experiment and (**D**) magnetic field control; (**E**) IR transmittance spectra of effluents obtained after the experiment; (**F**) the kinetics of the gaseous solute concentration (hexane) in the effluent estimated using an IR spectrometer in terms of volume of gas transported (**a**) in the absence of a magnetic field and (**b**) in the presence of a permanent magnet unit; (**G**) morphology of gas−liquid boundary during the mass transfer process. Reprinted with permission from Ref. [46]. Copyright 2020, Elsevier.

**Figure 5 molecules-27-07931-f005:**
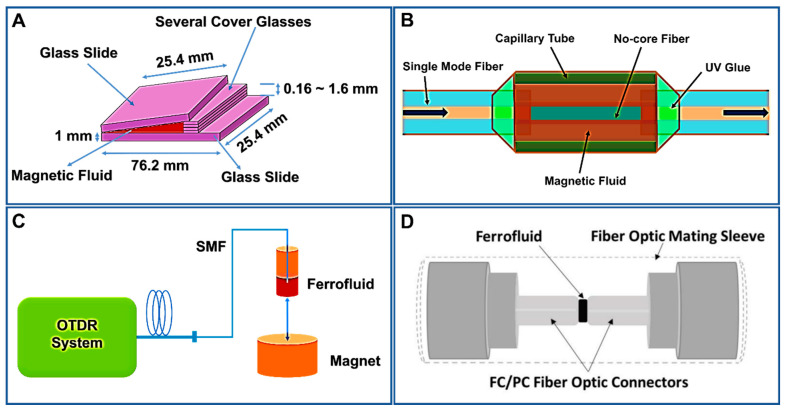
(**A**) Schematic illustration of the V−shaped groove. Adapted with permission from Ref. [69]. Copyright 2012, OPTICA Publishing Group; (**B**) schematic diagram of the FN−based magnetic field sensor. Adapted with permission from Ref. [71]. Copyright 2013, OPTICA Publishing Group; (**C**) schematic of the setup used for the magnetic field sensing. Adapted with permission from Ref. [73]. Copyright 2011, AIP Publishing; (**D**) schematic of the assembled extrinsic Fabry–Perot interferometer (EFPI) sensor. Reprinted with permission from Ref. [74]. Copyright 2014, MDPI.

**Figure 6 molecules-27-07931-f006:**
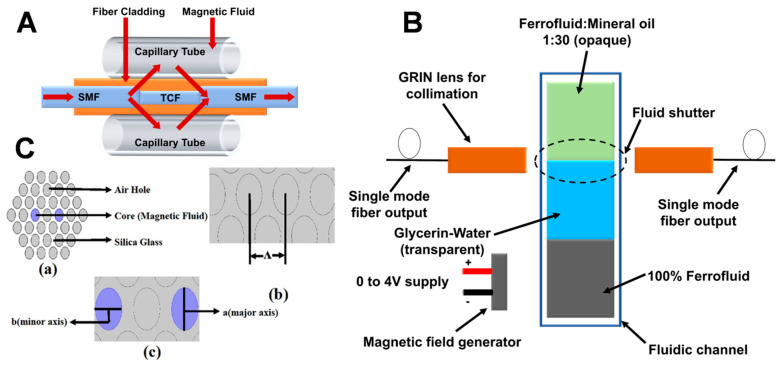
(**A**) A novel magnetic sensor employing a thin−core fiber (TCF) interferometer enclosed by ION−based FNs: standard modal interferometer; (**B**) schematic layout of ION FN−based optical fiber magnetic field sensor; (**C**) cross−sectional view of the novel magnetic field sensor using air holes and a silica cladding with Fe_3_O_4_ NP−based FNs. Reprinted with permission from Ref. [77]. Copyright 2020, Elsevier.

**Figure 7 molecules-27-07931-f007:**
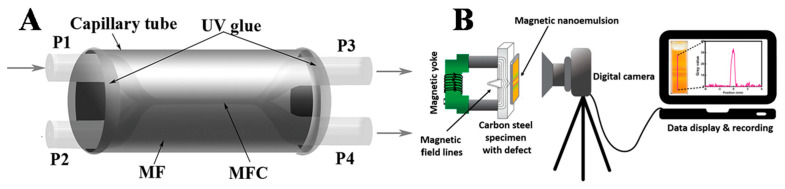
(**A**) Schematic of an MFC−based magnetic field sensor with an MF environment. Reprinted with permission from Ref. [78]. Copyright 2015, AIP Publishing; (**B**) schematic representation of the experimental setup for defect detection using MNE. Reprinted with permission from Ref. [81]. Copyright 2020, Elsevier.

**Figure 8 molecules-27-07931-f008:**
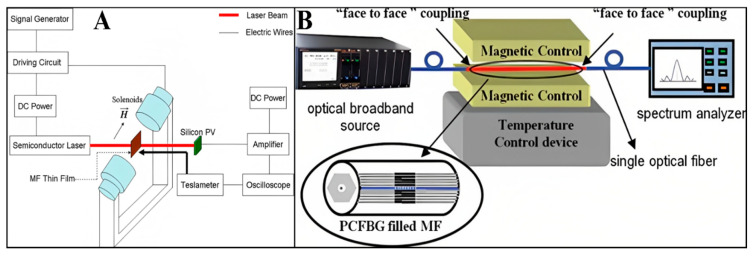
(**A**) Schematic diagram of experimental arrangement for realizing temperature sensor using an FN thin film. Reprinted with permission from Ref. [82]. Copyright 2008, Springer Nature (**B**) setup used for estimating magnetic field and temperature using PCFBG loaded with magnetic nanofluid. Reprinted with permission from Ref. [83]. Copyright 2013, Taylor & Francis.

**Figure 9 molecules-27-07931-f009:**
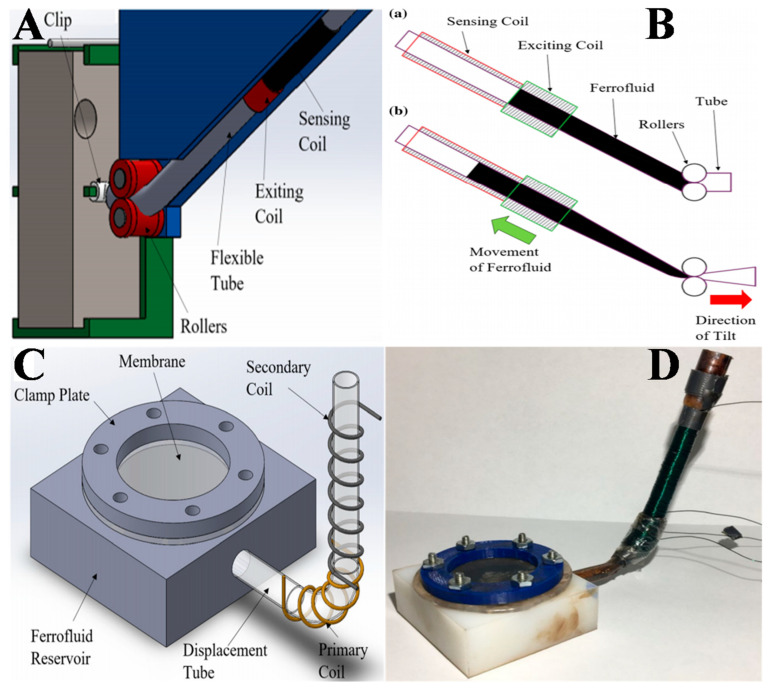
(**A**) Model of the proposed tilt sensor showing the connections of mass, coils, tubes, and rollers; (**B**) working principle of the proposed tilt sensor illustrating its roller system when the sensor is (**a**) “at rest” and (**b**) in a “tilting” state. Reprinted with permission from Ref. [93]. Copyright 2020, Springer Nature (**C**) Model of the FN−based force sensor; (**D**) fabrication of the proposed FN−based force sensor. Reprinted with permission from Ref. [94]. Copyright 2020, Springer Nature.

**Figure 10 molecules-27-07931-f010:**
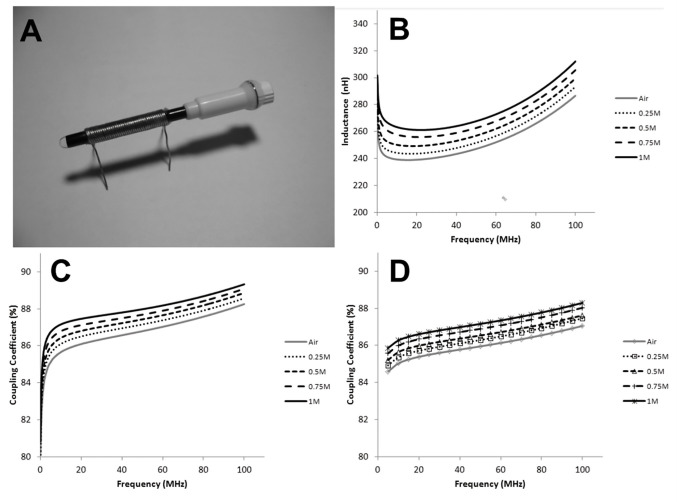
(**A**) Transformer on a capillary using oil−based ION FNs; (**B**) the inductances of transformer coils with different types of magnetic cores; the coupling coefficients according to (**C**) measured data and (**D**) simulated data of on−chip transformers with different magnetic cores (air and Fe_3_O_4_ NP−based FNs of 0.25, 0.5, 0.75, and 1 M). Reprinted with permission from Ref. [106]. Copyright 2011, Springer Open.

**Figure 11 molecules-27-07931-f011:**
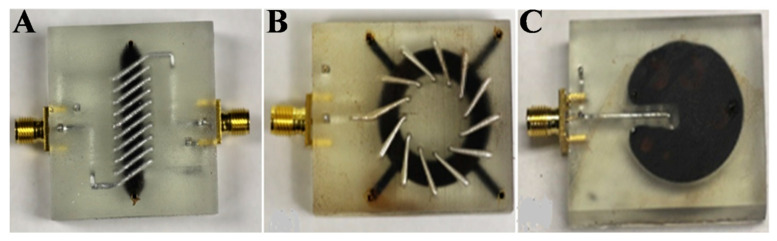
(**A**) FN-cored transformer; (**B**) FN−cored toroidal inductor; (**C**) planar coil−based inductive wireless power system with magnetic backplane using FN. Reprinted with permission from Ref. [107]. Copyright 2019, Elsevier.

**Figure 12 molecules-27-07931-f012:**
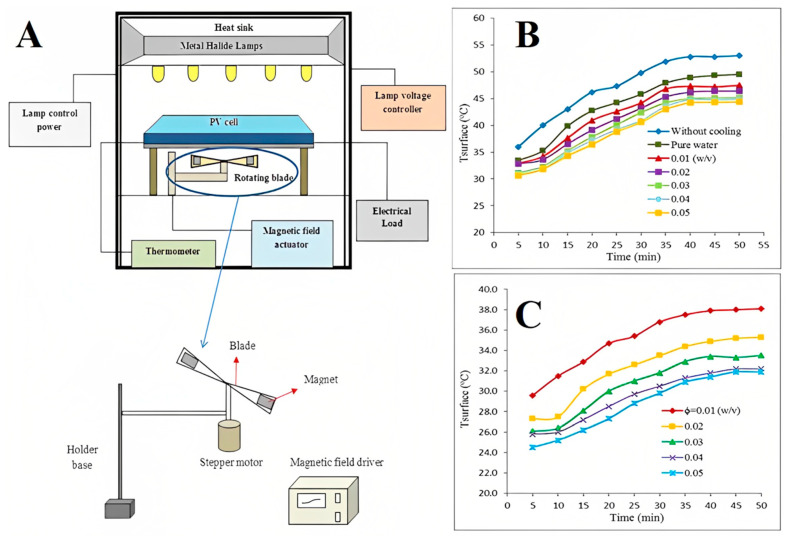
(**A**) Schematic diagram of experimental setup; variation of the average temperature of the PV cell surface overtime (**B**) without RMF and (**C**) with RMF with B=880 mT and ω=8 rad/s. Reprinted with permission from Ref. [116]. Copyright 2018, Elsevier.

**Figure 13 molecules-27-07931-f013:**
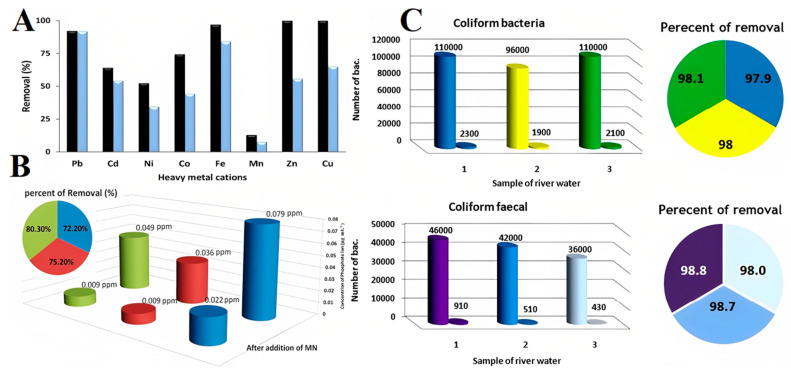
(**A**) Removal percentage of heavy-metal cations with MNPs at two different concentrations of heavy-metal cations and constant water volume (conditions: adsorbent, 0.15 g; volume, 200 mL; concentrations of heavy metal cations, 1 μg·mL^−1^ (black bar) and 4 μg·mL^−1^ (blue bar); temperature, 25 °C); (**B**) effect of MNPs on phosphate ion removal. Concentrations before and after treatment and removal efficiency are shown in the pie chart (volume of water, 100 mL; pH, 8.3); (**C**) antibacterial activity of MNPs in the presence of *coliform* bacteria and *coliform* fecal bacteria in real water samples (the number of bacteria before and after treatment, *n* = 3). Reprinted with permission from Ref. [136]. Copyright 2016, Elsevier.

**Figure 14 molecules-27-07931-f014:**
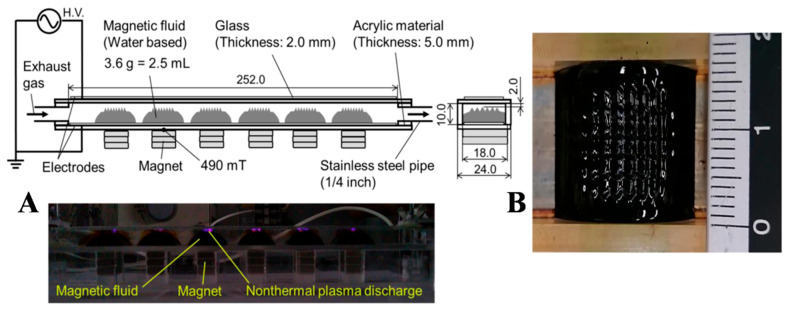
(**A**) Schematic of magnetic nanofluid filter and image of NTP discharge from magnetic nanofluid; (**B**) image of spikes in magnetic nanofluid with scale. Reprinted with permission from Ref. [140]. Copyright 2020, Elsevier.

**Figure 15 molecules-27-07931-f015:**
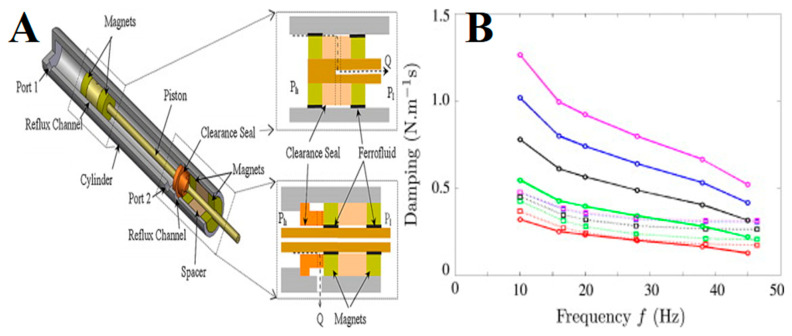
(**A**) Photograph of a practical implementation of an actuator consisting of two hybrid FN seals. Reprinted with permission from Ref. [150]. Copyright 2009, Elsevier; (**B**) Damping of APGW10 ferrofluidic seal as a function of volume (vol.-1: red; vol.-2: green; vol.-3: black; vol.-4: blue; vol.-5: magenta): experimental (dotted lines); model (continuous lines). Reprinted with permission from Ref. [154]. Copyright 2014, Elsevier.

**Table 1 molecules-27-07931-t001:** Specifications of ION-based FNs and their important advanced technological applications concerning engineering fields other than biomedical.

ION Size (nm)	Concentration	Amplitude of Mag. Field	Applications	Results	Ref.
15	0.39%	100–1000 G	Mass transfer	92.8% enhancement	[39]
13.2	0.3%	-	Mass transfer	164% enhancement	[40]
17	0.002 wt.%	-	Mass transfer	72% enhancement	[41]
17	0.001–0.005 wt.%	0.36–1.45 T	Mass transfer	121% enhancement	[43]
14	-	200×10^−6^ T	Magneto-optic sensor	Sensitivity of 276 mT·A^−1^	[67]
10	0.39 vol.%	2013 Oe (perpendicular) and 2167 Oe (parallel)	Magneto-optic sensor	Sensitivity 0.3998 dB/Oe (perpendicular) and 0.0017 dB/Oe (parallel)	[68]
10	-	0–220 Oe	Optical fiber magnetic field sensor	Sensitivities up to 905 pm/mT and 0.748 dB/mT	[71]
20	0.3–0.6 wt.%	0–80 mT	Magnetic field sensor	Higher sensitivity of 155.7 mT and 242 pm/mT for 0.3 wt.% and 0.6 wt.%, respectively	[73]
10	7.9 and 17.7 vol.%	−3000 to 3000 mG	Fiber-optic magnetic sensor	Higher sensitivity of 0.3 to 2.3 nm/mT	[74]
10	-	0–240 Oe	Magnetic field optical fiber sensor	Sensitivity of 11.8 pm/Oe	[75]
9	2.5%	25.28–489.56 mT	Magnetic field sensor	Sensitivity of 25.3 to 83.5 mT	[80]
10	6.47 vol.%	40.9 mT	Temperature sensor	Detected above 60 °C	[82]
-	3 vol.%	20–300 Oe	Temperature sensor	Detection limit ±0.82 °C	[83]
10–15	3 vol.%	-	Temperature sensor	Sensitivity 3.7 mK	[84]
-	-	-	Force sensor	68.3 mV/N	[94]
10	0.5–1 M	-	Transformer	Improved inductance and coupling efficiencies	[96]
<15	-	-	Transformer	Coupling improvements and nearly threefold increase in inductance density	[107]
15–20	1–3 vol.%	135–160 mT	Micropump device	Cell sorting devices (6.1 µL/min), low-flow drug delivery systems (1–10 µL/min), and pathogen detection systems (3–5.83 µL/min).	[109]
45	3 wt.%	300 G	Coolant	Overall cooling efficiency increased by 79%	[115]
25	0.05% (w/v)	880 mT	Coolant	Power (P) increased by 47.5% and thermal efficiency increased by 30%	[116]
30–50	-	-	Water purification	98% removal of *fecal coliform* bacteria	[136]
9.5–11.3	-	-	Water purification	86% efficiency of Co^2+^ adsorption	[139]
9.5–11.3	-		Water purification	30% efficiency of Co^2+^ adsorption	[139]
10	0.3 vol.%	490 mT	Air purification	PM removal efficiencies (*η*) of44% for PMs with diameters d_p_ > 0.3 μm, 85% for d_p_ > 0.5 μm, 99% for d_p_ > 1.0 μm, and 100% for d_p_ > 2.0 μm	[140]
-	-	-	Seal	Sealing capacity improved	[143]
	0.5 wt.%	-	Lubricant	19.4% reduction in friction coefficient	[144]
20	30 vol.%	10–650 mT	Brake	Braking torque increased by1300 N·m and 410 N·m for axial and radial squeezing stress, respectively.	[152]
-	-	1000 kA·m^−1^	Damper	Higher magnetic saturation and higher damping	[154]

## Data Availability

This manuscript has no associated data as no new datasets were generated or analyzed.

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
