# Peer review of "Iron Oxide Nanoparticle-Based Ferro-Nanofluids for Advanced Technological Applications"

_molecules, 2022, doi:10.3390/molecules27227931_

Round 1

Reviewer 1 Report

This review paper, namely “Iron Oxide Nanoparticles Based Ferro-Nanofluids for Advanced Technological Applications”. This study aimed to highlight current advancements in IONs-based FNs’ development and applicability beyond biomedical applications. As a result, a comprehensive overview of the recent improvement of FNs based on IONs for various improved engineering applications is unquestionably necessary. It is divided into four sections: (i) chemical engineering applications, (ii) encompasses electrical and electronics applications, (iii) environmental engineering applications (water and air purifications), and (iv) briefly discusses magneto-rheological applications (dampers and sealings). Lastly, the conclusion, challenges, and future perspectives of IONs-based FNs are discussed. It is a well-organized paper and informatics literature for readers.

Some suggestions are given below.

1.       It is suggested that section 3.4 requires references where necessary, specially Lines 625-626 “This was first of its kind fabricated through 3D printing technique.” Give references

2.       Please revise the conclusion section.

Author Response

Comment #1: It is suggested that section 3.4 requires references where necessary, specially Lines 625-626 “This was first of its kind fabricated through 3D printing technique.” Give references

Response: Thanks for your suggestion. Needful has been done.

Comment #2: Please revise the conclusion section.

Response: Thank you for your suggestion. The conclusion section has now been modified.

Reviewer 2 Report

This review article mainly discusses about the utilization of iron oxide nanoparticles based ferro-nanofluids for applications in major fields like chemical engineering, electronics/electrical engineering and etc. Though this work is extensive and the article is written interestingly, few changes needed to be done based on following comments before publication.

·         The beginning 4-5 sentences of the abstract are not clear and it can be modified.

·         The introduction section is elaborately written as it covers the basics of nano, iron, iron oxide, magnetic nanoparticles and their biomedical applications. As this information is widely available already, the “Introduction” part can be shortened.

·         A scheme specific to this review can be drawn and separately presented; as Figure 1 includes all applications, which may be confusing to the readers.

·         In some figures, the internal details are not visible.

Author Response

Comment #1: The beginning 4-5 sentences of the abstract are not clear and it can be modified.

Response: Thanks for your kind notice. The abstract section has now been modified.

Comment #2: The introduction section is elaborately written as it covers the basics of nano, iron, iron oxide, magnetic nanoparticles and their biomedical applications. As this information is widely available already, the “Introduction” part can be shortened.

Response: Thank you for your good suggestion.  The introduction section has now been modified considerably.

Comment #3: A scheme specific to this review can be drawn and separately presented; as Figure 1 includes all applications, which may be confusing to the readers.

Response: Thank you for your kind notice. Figure 1 has now been replaced with a relevant Figure.

Comment #4: In some figures, the internal details are not visible.

Response: Thank you for your kind notice. The Figures’ resolutions have now been enhanced.

Reviewer 3 Report

The authors presented the paper "Iron Oxide Nanoparticles Based Ferro-Nanofluids for Advanced Technological Applications"

1) I recommend to add some additional keywords (10 max) in the list about your applications. In the abstract you mention twice that you will not discuss biomedical application and will discuss engineering applications. Please do it only once. What is MEMS? Decrypt it.

2) The area of magnetic nanoparticles is developing very fast. In this way, I recommend to update the reference list. Do not use too old paper (older than 5 years). Much more fresh 2-3 years paper should be presented. There are many 2022 year papers.

3) I am a bit confused. In the abstract you write that you want present "applicability beyond biomedical applications". However, in the text and Figures, for example Fig., MRI, drug delivery, and other biomedical things. Please, change the abstract or reduce clinical information application discussion in the abstract. As you known, there are a number of reviews with biomedical applications. I recommend focusing on not biomedical applications in the Introduction and the Conclusion sections, and other parts. It is much newer.

Minor comment

Fig. 8b bad resolution

Table 1.It will be good to expand and shrink part of the columns to present this Table on one page. Moreover, you can delete Fe3O4 column.

Author Response

Comment #1: I recommend to add some additional keywords (10 max) in the list about your applications. In the abstract you mention twice that you will not discuss biomedical application and will discuss engineering applications. Please do it only once. What is MEMS? Decrypt it.

Response: Thank you for your important suggestions. Additional keywords have now been added. Also, the abstract section has now been modified suitably and the full form of MEMS is mentioned.

Comment #2: The area of magnetic nanoparticles is developing very fast. In this way, I recommend to update the reference list. Do not use too old paper (older than 5 years). Much more fresh 2-3 years paper should be presented. There are many 2022 year papers.

Response: Thank you for your good suggestion. Since the introduction section has been modified considerably, a number of old references have now been removed.

Comment #3: I am a bit confused. In the abstract you write that you want present "applicability beyond biomedical applications". However, in the text and Figures, for example Fig., MRI, drug delivery, and other biomedical things. Please, change the abstract or reduce clinical information application discussion in the abstract. As you know, there are a number of reviews with biomedical applications. I recommend focusing on not biomedical applications in the Introduction and the Conclusion sections, and other parts. It is much newer.

Response: Thank you for your suggestion. The abstract, introduction and conclusion sections have now been modified suitably. Also, Figure 1 has now been replaced with a relevant Figure.

Comment #4:  Fig. 8b bad resolution

Response: Thank you for your suggestion. The resolution of Fig.8B has now been enhanced.

Comment #5: Table 1. It will be good to expand and shrink part of the columns to present this Table on one page. Moreover, you can delete Fe3O4 column.

Response: Thank you for your suggestion. Table 1 has now been modified and the Fe3O4 column is also removed.